# Relationship between Vitamin Deficiencies and Co-Occurring Symptoms in Autism Spectrum Disorder

**DOI:** 10.3390/medicina56050245

**Published:** 2020-05-20

**Authors:** Madalina-Andreea Robea, Alina-Costina Luca, Alin Ciobica

**Affiliations:** 1Department of Biology, Faculty of Biology, “Alexandru Ioan Cuza” University of Iasi, 700505 Iasi, Romania; madalina.robea11@gmail.com; 2Department of Pediatric Cardiology, Faculty of Medicine, ”Grigore T. Popa” University of Medicine and Pharmacy, 700505 Iasi, Romania; 3Department of Research, Faculty of Biology, “Alexandru Ioan Cuza” University of Iasi, 700505 Iasi, Romania

**Keywords:** autism spectrum disorder, nutrition, vitamins deficiencies

## Abstract

Recently, connections have been made between feeding and eating problems and autism spectrum disorder (ASD) and between autism pathophysiology and diet issues. These could explain some of the mechanisms which have not yet been discovered or are not sufficiently characterized. Moreover, there is an increased awareness for micronutrients in ASD due to the presence of gastrointestinal (GI) problems that can be related to feeding issues. For example, levels of vitamins B_1_, B_6_, B_12_, A and D are often reported to be low in ASD children. Thus, in the present mini review we focused on describing the impact of some vitamins deficiencies and their relevance in ASD patients.

## 1. Introduction

Autism spectrum disorder is a neurodevelopmental disorder which appears early in childhood, and it can cause social interaction deficits, communication impairments and behavioral challenges [1]. Over the years, scientific research had as a main purpose finding the causes of ASD appearance. Unfortunately, nowadays it is still unknown and the prevalence of this disease grew with an amazing speed among children. A recent report added new insights regarding ASD prevalence for 2016 in United States, stating that one in 54 eight-year-olds was diagnosed with autism, according to Centers for Disease Control and Prevention [2]. The symptoms are multiple starting with social deficits, repetitive behaviors, stress, sleep problems and also eating and feeding problems. Each individual is different from another and the symptoms may vary.

Feeding problems are common in ASD and they have been overlooked for a long time in favor of behavioral, social and communication deficits [3]. These problems are generally reflected through a selective texture of the food, a particular presentation or even preference for a certain meal [3,4,5]. Because of this situation there are differences between children with ASD and normal children. ASD children tend to gain weight more quickly compared with the general population [3]. In one study performed by Hill et al. (2015) it was confirmed that being overweight or obese is present in ASD children more than in normal ones. Also, they found evidence that suggest the existence of multiple factors associated with autism, such as sleep and affective problems, lower parent education, economic status or even location [3].

In the present mini review, we summarize several reports that have studied the correlation between ASD and eating and feeding problems. In addition, we present here the deficiency of vitamins in the human body which seemingly could lead in some cases to a dysregulated functioning of the nutritional status of ASD individuals.

## 2. What is Autism Spectrum Disorder?

Autism spectrum disorder is the name for a series of behavioral and developmental issues that appear in children [6] The term ”autism” was used by Paul Eugene Bleuler to define schizophrenia symptoms for the first time in 1912, but, several decades later, Hans Asperger and Leo Kanner shed light on this and put forward the modern basis of autism [7,8]. Over the years, under the name of autism spectrum disorder were reunited numerous disorders which had in common the main core of autism [8,9]. The American Psychiatric Association’s Diagnostic published a new version of the Diagnostic and Statistical Manual of Mental Disorder (DSM), fifth edition in 2013, in which the ASD diagnosis was changed [7]. Persistent deficits in social communication/ interaction and restricted, repetitive patterns of behaviors, interests, or activity are the criteria of ASD diagnosis [10]. Since the appearance of DSM-5, ASD severity is measured based on level of support needed: level 1 (requiring support), level 2 (requiring substantial support) and level 3 (requiring very substantial support) to describe the status of an ASD individual [8]. This change was made due to the fact that there were some differences between Asperger Syndrome (AS), Autistic Disorder (AD) and Pervasive Developmental Disorder – Not Otherwise Specified (PDD-NOS) but, at the same time, according to genetic analysis, similarities between these were high, favoring the DSM-5 criteria [8].

Before the appearance of DSM-5, ASD was diagnosed clinically based on three domains of impairment: social communication, interaction, and restricted repetitive and stereotyped behavior. Now, individuals with ASD must fulfill two main domains: social communication and interaction, and restricted repetitive behavior [10]. Furthermore, in DSM-5 are presented three levels of severity, as mentioned above, which should be used in ASD identification and diagnosis Figure 1. For the first level, named ”requiring support”, the key points for which an individual can receive this level are specified as: the incapacity to speak in public places even if in a familiar place he can speak normally, difficulty to start a conversation or to change the activities between them [9].

Moreover, besides DSM-5, there are another tests used for screening and diagnosis of autism as: Childhood Autism Rating Scale (CARS), Aberrant Behavior Checklist (ABC), Social Responsiveness Scale (SRS), Pervasive Development Disorder Behavior Inventory (PDD-BI), Autism Evaluation Treatment Checklist (ATEC), Severity of Autism Scale (SAS), Autism Diagnostic Observation Scale (ADOS), Autism Diagnostic Interview-Revised (ADI-R) and Clinical Global Impressions-Improvement (CGI-I) [11].

Regarding the etiology, ASD is a multifactorial disorder resulting from the combined action of genetic and environmental factors. Studies made on family members and twins showed that ASD is a heritable brain disorder beside genetic variations [10,12,13,14,15]. In close connection with genetic risks, environmental factors as drugs, prenatal viral infection, prenatal and perinatal stress, toxins, advanced parental age, vitamins and minerals deficiency can contribute to ASD appearance [1,7,10,16,17,18]. Due to the complexity and severity of symptoms, it is important to do further research to improve the clinical features of ASD.

## 3. Feeding and Eating Problems in ASD Individuals

ASD can be identified through symptoms grouped into two categories: the core and the secondary symptoms [19]. Feeding and eating problems can be found under the umbrella of secondary symptoms. Atypical eating behaviors and feeding problems are often reported by the parents of a child with ASD [20,21]. Food refusal, preferences for a certain product or foods, an obsessive routine for taking meals, and preference for the color and the texture of a specific kind of food are the most common [4,21,22,23,24]. Due to this fact, children and adults with ASD have deficiencies regarding the nutrients intake Figure 2, especially micronutrients [17].

As previously mentioned, food selectivity is a common symptom in ASD. Children who are diagnosed with ASD have preferences for snack foods, processed foods, starches, and less interest for vegetables, fruits and proteins [11,20,22]. Because of this behavior, there is a high prevalence of obesity among ASD children, according to several reports [3,23,25]. Recently, Kamal et al. (2019) published a study in which 151 children (2–18 years) with ASD were analyzed at the Child Development Center at University Kebangsaan Malaysia Medical Center. Their results showed that older children tend to gain weight, while the youngest are underweight. The age, the high maternal BMI, the absence of physical activity, food refusal and preference are the main factors which promote the obesity of ASD children [25].

Many individuals with ASD have been shown to experience a series of gastrointestinal (GI) issues, such as: constipation, stomach pain, nausea, diarrhea, abdominal pain, blood in stools and vomit [20,26,27,28]. The reported prevalence of GI symptoms in ASD individuals has ranged from 9% to 90% in 2010; most of the sources had at least one specific symptom of GI dysfunction [26]. A recent published study (2019) with 340 ASD participants reported that 65% of them had constipation, 47.9% stomach pain, 23.2% nausea and 29.7% diarrhea [28]. The strong correlation of gastrointestinal symptoms with ASD severity is not completely resolved; there are conflicting findings, where diet and medication are not linked with GI dysfunction, for example [29]. Thus, the GI tract is responsible with nutrient absorption and disruptions in this process can lead to severe deficiencies of micronutrients and macronutrients [30].

Despite these consequences, there are some treatment options for feeding and eating problems for autistic people. The effectiveness of a certain treatment depends by the severity of feeding problems. As main interventions in these situations, escape extinction, differential reinforcement, the Premack principle, and manipulation of food texture or blending a preferred and non-preferred food together have been used [31,32,33]. Even if these were demonstrated through empirical studies, it proved that those methods could be combined in feeding and eating problems in ASD [32,34,35].

## 4. Perturbation of Micronutrients Function in ASD Children

Micronutrients are vitamins and minerals which can be found in small amounts in the human body, and their absence leads to a perturbation in enzyme, hormone or other substance production; they are essential for development and maintenance of a normal body functioning [36]. Studies have reported numerous cases of ASD individuals who presented low levels of vitamins and minerals [1,37,38]. Moreover, it was confirmed that administration of vitamins during the first month of pregnancy could prevent or reduce the recurrence in siblings of children with ASD [39,40]. A list with vitamins, effects after vitamin supplementation or side effects of it are summarized in Table 1.

### 4.1. Thiamine (Vitamin B_1_)

Thiamine was the first B vitamin discovered, and was synthesized in 1936 by Robert Williams [47]. Vitamin B_1_ (vit. B_1_) is a water-soluble vitamin and it has been correlated with many CNS diseases [1,48,49]. Thiamine deficiency was known as ”beriberi” for a long time and, in Japan it was named “national disease” due to the high prevalence throughout the population [47]. It was, also, linked with Alzheimer’s disease (AD) after the appearance of plaques and β-amiloid in animal models [50]. The implication of thiamine in ASD can be seen via its effects on apoptotic factors (factor p-53, caspase-3, and Bcl-2), oxidative stress (prostaglandins, reactive oxygen species, nitric oxide synthase, mitochondrial dysfunction) and neurotransmitter systems (serotonin, glutamate, and acetylcoline) [51].

One pilot study investigated the effect of thiamine tetrahydrofurfuyl disulfide (TTFD) on a group formed by 10 autistic children. Children received a dose of 50 mg TTFD twice a day for two months. E2 and ATEC forms were used to check the changes. After two months of vitamin supplementation, eight of the 10 children improved clinically [41]. Several years later, Adams et al. (2011) studied whether or not 20 mg of thiamine had positive effects on a group formed by 141 children with autism after three weeks of administration. For assessing the autistic symptoms and severity, three tests were used at the beginning and end of the study: PDD-BI, ATEC and SAS. The results were also positive [42].

Quantitation of plasma thiamine and its metabolites was measured in 27 ASD children. All subjects received the ASD diagnosis after consulting the Child Neurological and Psychiatric Unit of the Bellaria Hospital of Bologna, DSM-5, ADOS and CARS tools. No deficiency of thiamine and thiamine monophosphate was found [52].

### 4.2. Pyridoxine (Vitamin B_6_)

Vitamin B_6_ (vit. B_6_) is a very important cofactor implicated in multiple biochemical reactions which has as main target the general metabolism of the cells [53]. It is a water-soluble vitamin and it is implicated in the synthesis, conversion and degradation of amino acids, fatty acids, and neurotransmitters. Gamma-aminobutyric acid (GABA), dopamine, noradrenaline, histamine, serotonin, glycine and D-serine need vit. B_6_ as coenzyme in their synthesis [53,54]. Children with ASD often report high deficiency in vit. B_6_. In several studies the level of pyridoxine before and after vitamin supplementation in ASD individuals were tested. Behavioral improvements were obtained after giving a combined dose of vit. B_6_ with magnesium (Mg) [42,43].

Positive results were obtained in a study performed in 2006 on 33 autistic children by Mousain-Bosc et al. Their study focused on the combined effect of vit. B_6_ and Mg. They gave 0.6 mg/kg/d vit. B_6_, the recommended dietary allowance (RDA), mixed up with 6 mg/kg/d Mg for six months. Intraerythrocyte Mg^2+^, serum Mg^2+^ and blood ionized Ca^2+^ were measured at the beginning and end of the treatment. Also, clinical symptoms were scored 0 to 4. Social interactions (23/33 children), restricted behavior (18/33), communication (24/33 children) and delayed functioning (17/33 children) were improved after this period [43].

Vit. B_6_ level is reported to be high in ASD children and this was interpreted as a consequence of insufficient pyridoxal kinase, an enzyme required for B_6_ metabolism [55]. This fact was showed in a control trial with ASD children where abnormally high plasma levels of vit. B_6_ were recorded compared to those from controls [37]. A randomized, double-blind, placebo-controlled vitamins treatment study was performed on 141 ASD children and adults for three months. Beside other vitamins, they received 40 mg of vit. B_6_. The result of this study concluded that vitamins supplementation can improve or normalize the gluthatione, plasma adenosine triphosphate (ATP), nicotinamide adenine dinucleotide (NADH), nicotinamide adenine dinucleotide phosphate (NADPH), plasma sulfate, biotin levels and decrease the oxidative stress biomarkers in ASD children [54]. Many studies of Adams et al. brought new perspectives regarding vitamins administration. They designed and conducted studies on people with ASD providing nutritional and metabolic information, which were completed with significant improvements [56].

The combined action of vit. B_6_ and Mg on stress status was studied in a single-blind clinical trial. People with low magnesemia showed reduced stress level after intake of 30 mg vit. B_6_ with 300 mg after eight weeks’ treatment [57].

### 4.3. Cobalamin (Vitamin B_12_)

Vitamin B_12_ (vit. B_12_) is an essential component for DNA synthesis and for production of cellular energy [1,58]. The deficiency of it can be observed in a series of symptoms specific to gastrointestinal, hematological, neurological and psychiatric disorders [59,60]. Neurological impairments, such as motor disturbances, abnormal balance and reflexes, sensory and memory loss, cognitive impairment, irritability, and cerebral atrophy, are some of the known features of cobalamin deficiency [61,62]. Furthermore, vit. B_12_ is implicated in the process of methylation and in the redox status. These are risks factors for ASD and there were reported some changes in plasma methionine, homocysteine, cysteine and glutathione (GSH) beside the nitrous oxide effect after vitamin intake [63,64].

Supplementation with vit. B_12_ has a beneficial effect on the methylation process, the potential of antioxidants and on GSH neurotransmitter, according to Hendren et al. (2016). The ASD symptoms were verified through ADOS and ADI-R. After eight weeks of injections with vit. B_12_ ASD children showed better responses in overall clinical-rated symptoms sustained by CGI-I score. Regarding the behavioral symptoms of ASD, these could not be improved by vit. B_12_ administration, as ABC and SRS showed [65].

Bertoglio et al. (2010) studied the effect of vit. B_12_ on a group formed by 30 ASD individuals for 12 weeks. They injected 0.06 mg/kg vit. B_12_ every three days. After three months of treatment, only nine individuals showed significant changes in behavior and GSH plasma levels as CGI-I and blood analysis revealed [44].

### 4.4. Vitamin A

Known as a fat-soluble vitamin, its role in human body is essential [66]. Vitamin A (vit. A) can be found under two forms: retinol, which is derived from animal sources, and provitamin A in fruits and vegetables [66]. The deficiency of vit. A is a public health problem due to its negative effects, such as anemia, weak resistance to infections, ocular symptoms, and the risk of death [67,68,69]. Many reports have demonstrated that children with ASD have numerous nutritional deficiencies, including lack of vit. A [38,70]. It has been reported that therapy with vit. A can improve the symptoms of ASD [70,71]. The severity of ASD was evaluated using CARS score. Dietary, feeding behavior and gastrointestinal questionnaires were performed and the levels of vit. B_12_, vit. A, vit. D, serum ferritin, folate and 25(OH) were measured. A study focused on ASD children from Chongqing (China) whose restricted diet, inadequate nutrient intake and mealtime behavioral issues favored vit. A deficiency, but it was negative correlated with CARS score according to Liu et al. (2016) [70].

In a recent study, a group of 33 ASD children from a pilot study showed that 0.06 mg/kg vit. A can improve autistic symptoms after measuring it through CARS score, reduce the blood serotonin level and produce significant changes in mRNA expression of RAR α, RAR β and TpH 1 [72]. A new report brings additional informational about vit. A and vit. D deficiencies in 332 ASD children. The data collected by laboratory and anthropometric measurements, Autism Behavior Checklist, CARS and SRS demonstrated that vit. A and vit. D deficiencies may play an important role in ASD worsening [73].

### 4.5. Vitamin D

Vitamin D (Vit. D) has gained a lot of interest in the recent years when some reports were sustaining its important action in bone metabolism [74]. Low levels of vit. D were reported in several studies in which ASD individuals were analyzed [20,75,76,77]. ASD individuals who received vit. D as an alternative treatment showed significant improvements [45,78,79]. In a study performed by Saad et al. (2016), the level of vit. D in ASD children was analyzed. According to their results, 57% of the individuals had vit. D deficiency and 30% had vit. D insufficiency. Furthermore, they had administered vit. D_3_ to ASD children in a dose of 0.0075 mg/kg/day for three months in an open-label trial. Almost 81% had positive results after this period [45]. The efficiency of vit. D supplementation is still sustained by studies recently made [46].

Changes in irritability and hyperactivity were reported in Mazahery et al.’s (2019) study. They administered vit. D_3_ through two ways: capsules with only 0.049 mg/kg/day vit. D_3_ and capsules with vit. D_3_ (0.049 mg/kg/day) and omega-3 LCPUFA (722 mg docosahexaenoic acid) for one year. The study revealed that vit. D_3_ alone or mixed with another supplement can improve the core of ASD children as ABC results showed, and even more, irritability and hyperactivity was significantly reduced [46].

## 5. Conclusions

Many studies in the field of autism have focused on elucidating the adequate interventions for treating ASD. Known as a neurodevelopmental disorder, ASD is diagnosed, mainly, through DSM-5, but the Childhood Autism Rating Scale, Aberrant Behavior Checklist and Social Responsiveness Scale are also used. Due to its multifactorial disorder, ASD treatment depends on the need of each individual. Thus, in ameliorating the ASD symptoms, vitamin supplementations are recommended as alternative therapy. In this mini review, we have gathered several studies in which the effects of vitamin B_1_, B_6_, B_12_, A and D intake on autistic individuals were highlighted. Beside these aforementioned vitamins, there are also other vitamins or combinations between micro- and macronutrients which need to be explored. Despite the positive effects recorded for vitamin intake, recommended supplementation is required from medical specialists. In conclusion, further research is needed to provide safety assurance and evidence of efficacy in vitamins administration. 

## Figures and Tables

**Figure 1 medicina-56-00245-f001:**
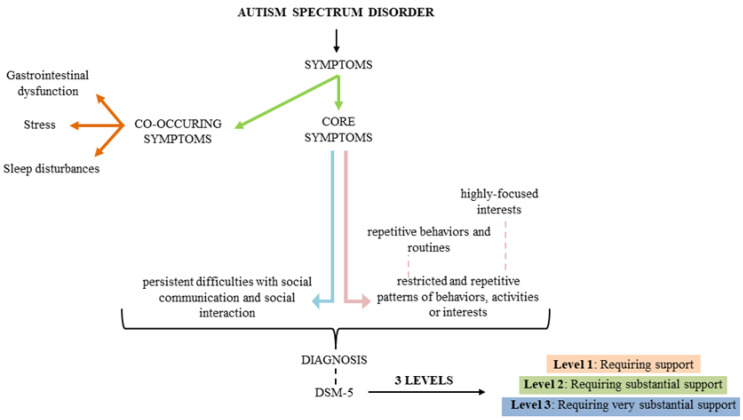
Characteristics of autism spectrum disorder (ASD).

**Figure 2 medicina-56-00245-f002:**
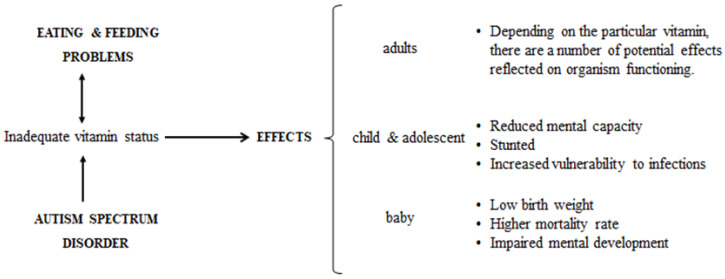
Effects of inadequate vitamin intake on ASD individuals.

**Table 1 medicina-56-00245-t001:** Vitamins used in ASD studies in relation to treatment efficiency.

Vitamin	Participants	Dosage	Time of Administration	Type of the Study	Effects	Side Effects	Observations	References
B_1_	10 children	50 mg TTFD	Twice a day / 2 months	PS	P.E.	-	↑clinical symptoms ↑Pb, Hg, Cd, As levels	[41]
	141 children	20 mg	Days 1–4: 1/6 of fd Days 5–8: 2/6 of fd Days 9–12: 3/6 of fd Days 13–16: 4/6 of fd Days 17–20: 5/6 of fd Days 21 and later: fd	RCT, DB, P	P.E.	diarrhea, constipation	↑ATP, sulfation, NADH, NADPH, GSH ↑clinical symptoms ↓ OS	[42]
B_6_	33 children	6 mg/kg Mg and 0.6 mg/kg vit. B_6_	6 months	RCT	P.E.	-	↑social interactions, communication, abnormal functioning, stereotyped restricted behavior ↑Erc-Mg values	[43]
	141 children	40 mg	Days 1–4: 1/6 of fd Days 5–8: 2/6 of fd Days 9–12: 3/6 of fd Days 13–16: 4/6 of fd Days 17–20: 5/6 of fd Days 21 and later: fd	RCT, DB, P	P.E.	diarrhea, constipation	↑ATP, sulfation, NADH, NADPH ↑clinical symptoms ↓ OS	[42]
B_12_	30 children	0.06 mg/kg	12 weeks	PS, DB	N.E.	-	↓ OS ↑GSH no behavioral improvements	[44]
D	122 children	0.0075 mg/kg	3 months	RCT	P.E.	skin rashes, diarrhea and itching	↑core symptoms	[45]
	73 children	0.049 mg/kg	1 year	RCT	P.E.	-	↓ irritability and hyperactivity	[46]

↑ rise/ improvement; ↓ decrease; As: arsenic; ATP: plasma adenosine-5’-triphosphate; Cd: cadmium; DB: double-blind; Erc-Mg: intraerythrocyte Mg^2+^; fd: full dose; GSH: glutathione; Hg: mercury; Mg: magnesium; N.E.: no effects; NADH: nicotinamide adenine dinucleotide; NADPH: nicotinamide adenine dinucleotide phosphate; OS: oxidative stress; P.E.: positive effects; P: placebo; Pb: lead; PS: pilot study; RCT: randomized controlled trial; TTFD: thiamine tetrahydrofurfuryl.

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
