# Peer review of "Relationship between Vitamin Deficiencies and Co-Occurring Symptoms in Autism Spectrum Disorder"

_medicina, 2020, doi:10.3390/medicina56050245_

Round 1

Reviewer 1 Report

Thank you for asking me to review the manuscript titled, “The Relevance of Vitamins in Autism Spectrum Disorder.” The following are my comments, section-by-section. In general, there are many instances of grammatical errors throughout the manuscript that I believe can be addressed through a revision with the editorial team. However, these errors usually do not affect the meaning of the points being made and are not too concerning.

Title

  • I think the title could better describe the review. One suggestion is “Relationship between Vitamin Deficiencies and Core- and Co-Occurring Symptoms in Autism Spectrum Disorder.”

Abstract

  • Would use ASD to refer to autism after the first use of autism spectrum disorder (ASD) on line 14.
  • Line 17 - would change to “For example, levels of vitamins B1, B6, B12, A, and D are often reported to be low in children with ASD.”
  • Line 16 - should be “sufficiently”

Introduction

  • Line 23 - ASD does not affect children predominantly, however, it is typically diagnosed in childhood. For this, please cite the DSM-V definition of ASD and use the language contained therein.
  • Line 16 - would not refer to ASD as a disease, but rather a disorder or a condition. Softer language is needed here.
  • I suggest adding that many children with ASD have gastrointestinal problems which may be related to their feeding issues.

Section 2

  • Line 45 - ASD appears early in childhood but it is not just confined to childhood.
  • I recommend citing the primary source for the definition of ASD in the DSM-V.
  • Also, please define ASD in the beginning, then use ASD for future mentions of autism throughout the rest of the manuscript.
  • Line 52 - please change to “DSM” instead of “DMS”
  • Line 66- this section is good!
  • Line 72 - since Figure 1 largely deals with the effects of vitamins in ASD, there needs to be a sentence here talking about this notion. This way, Figure 1 will make more sense to the reader by setting them up for the rest of the paper.
  • For Figure 1, “SYMPTOMS” in the middle of the figure should probably read “Co-occurring Symptoms” as GI and sleep problems are not considered to be core symptoms. Then, have another arrow pointing to core symptoms (social communication , repetitive behaviors, etc.)

Section 3

  • Might consider mentioning gastrointestinal issues here as they are prevalent in ASD (e.g, Ferguson et al recently found that 65% of children and adolescents with ASD had co-occurring constipation; earlier, Buie et al. found a range of 9-90% of patients with ASD had at least 1 GI disorder).
  • Further, perhaps GI disorders in ASD might impact absorption of micro and macronutrients?

Section 4

  • Table 1 - please define TTFD below the table so it is clear to the reader who quickly glances at the table.
  • It may help to list in Table 1 if each study is an RCT or if it’s open label
    • Also, for effects, can you list a brief description of what the effects were? For example, for increases in sociability, put an up arrow followed by sociability. Also, if there were notable side effects, it would be helpful to list this in Table 1.
    • Further, the actual positive effects should be listed in the text.
  • Line 129 - spelling should be “noradrenaline”
  • In some places (e.g., line 130) vit. Is used, but in others (line 132) vitamin is used. Please be consistent here and throughout the manuscript.

Section 5

  • The conclusion section could benefit from some additional points. Were the studies RCTs or unblinded? If so, perhaps a call for more RCTs should be mentioned.
  • Also, any negative side effects of vitamin supplementation?
  • This is a mini review, so mention that the small list of vitamins isn’t exhaustive, and that others may provide benefit.
  • Further, combinations of vitamins should be explored, not just increasing one vitamin at a time.
  • A brief statement that individuals seeking to supplement vitamins should consult with their physician seems appropriate.

Overall, this mini review is good but would benefit from addressing the concerns above. I congratulate the authors on their work and concern for those with ASD and wish them all the best with their future endeavors.

Author Response

  • I think the title could better describe the review. One suggestion is “Relationship between Vitamin Deficiencies and Core- and Co-Occurring Symptoms in Autism Spectrum Disorder.”

This was a good suggestion and we appreciated it, so we changed our title.

  • Would use ASD to refer to autism after the first use of autism spectrum disorder (ASD) on line 14. Done.
  • Line 17 - would change to “For example, levels of vitamins B1, B6, B12, A, and D are often reported to be low in children with ASD.” Done.
  • Line 16 - should be “sufficiently” . Done.
  • Line 23 - ASD does not affect children predominantly, however, it is typically diagnosed in childhood. For this, please cite the DSM-V definition of ASD and use the language contained therein. Done.
  • Line 16 - would not refer to ASD as a disease, but rather a disorder or a condition. Softer language is needed here. I suggest adding that many children with ASD have gastrointestinal problems which may be related to their feeding issues.

We reformulated the sentence and we added this part which the reviewer suggested.

  • Line 45 - ASD appears early in childhood but it is not just confined to childhood.

We changed the meaning sentence as it was indicated.

  • I recommend citing the primary source for the definition of ASD in the DSM-V. Also, please define ASD in the beginning, then use ASD for future mentions of autism throughout the rest of the manuscript. Done.
  • Line 52 - please change to “DSM” instead of “DMS”. Done.
  • Line 66- this section is good!
  • Line 72 - since Figure 1 largely deals with the effects of vitamins in ASD, there needs to be a sentence here talking about this notion. This way, Figure 1 will make more sense to the reader by setting them up for the rest of the paper.

For Figure 1, “SYMPTOMS” in the middle of the figure should probably read “Co-occurring Symptoms” as GI and sleep problems are not considered to be core symptoms. Then, have another arrow pointing to core symptoms (social communication , repetitive behaviors, etc.).

We made the adjustements for Figure 1 as the reviewer #2 indicated; we have now two figures instead of one. Moreover, we divided ”symptoms” în core and co-occurring symptoms as it was suggested.

  • Might consider mentioning gastrointestinal issues here as they are prevalent in ASD (e.g, Ferguson et al recently found that 65% of children and adolescents with ASD had co-occurring constipation; earlier, Buie et al. found a range of 9-90% of patients with ASD had at least 1 GI disorder).

Further, perhaps GI disorders in ASD might impact absorption of micro and macronutrients?

We added new reports for GI problems and, of course, these mentioned by the reviewer. In addition, we described what happens when absorption of micro and macronutrients is perturbed as GI disorders consequence.

  • Table 1 - please define TTFD below the table so it is clear to the reader who quickly glances at the table. It may help to list in Table 1 if each study is an RCT or if it’s open label Also, for effects, can you list a brief description of what the effects were? For example, for increases in sociability, put an up arrow followed by sociability. Also, if there were notable side effects, it would be helpful to list this in Table 1. Further, the actual positive effects should be listed in the text.

The table was updated as the reviewer indicated with 2 columns (the type of the study and observations).

  • Line 129 - spelling should be “noradrenaline”. Done.

  • In some places (e.g., line 130) vit. Is used, but in others (line 132) vitamin is used. Please be consistent here and throughout the manuscript. Checked.

  • The conclusion section could benefit from some additional points. Were the studies RCTs or unblinded? If so, perhaps a call for more RCTs should be mentioned. Also, any negative side effects of vitamin supplementation?

We mentioned in the table where the side effects were recorded.

  • This is a mini review, so mention that the small list of vitamins isn’t exhaustive, and that others may provide benefit. Further, combinations of vitamins should be explored, not just increasing one vitamin at a time. A brief statement that individuals seeking to supplement vitamins should consult with their physician seems appropriate.

The conclusion paragraph was reformulated with suggestions above mentioned by the reviewer.

Moreover, we added several new reports through the review:

[1]          P. M. . Maenner, M. J.; Shaw, K. A.; Baio, J.; Washington, A.; Patrick, M.; DiRienzo, M.; Christensen, D. L.; Wiggins, L. D.; Pettygrove, S.; Andrews, J. G.; Lopez, M.; Hudson, A.; Baroud, T.; Schwenk, T.; White, T.; Rosenberg, C. R.; Lee, L.-C.; Harrington, R. A, “Prevalence of Autism Spectrum Disorder Among Children Aged 8 Years — Autism and Developmental Disabilities Monitoring Network, 11 Sites, United States, 2016,” MMWR Surveill Summ, no. SS-4, pp. 1–12, 2020, doi: 10.15585/mmwr.ss6904a1.

[3]          S. L. Hyman, S. E. Levy, and S. M. Myers, “Identification, Evaluation, and Management of Children With Autism Spectrum Disorder,” Pediatrics, vol. 145, no. 1, p. e20193447, Jan. 2020, doi: 10.1542/peds.2019-3447.

[4]          M. . Marotta, R.; Risolei, M. C., Messina, G., Parisi, L.; Carotenuto, M.; Vetri, L.; Roccella, “The Neurochemistry of autism,” Brain Sci., vol. 10, no. 163, 2020, doi: 10.3390/brainsci10030163.

[5]          T. Buie et al., “Evaluation, diagnosis, and treatment of gastrointestinal disorders in individuals with ASDs: a consensus report.,” Pediatrics, vol. 125 Suppl, pp. S1-18, Jan. 2010, doi: 10.1542/peds.2009-1878C.

[6]          V. Chaidez, R. L. Hansen, and I. Hertz-Picciotto, “Gastrointestinal problems in children with autism, developmental delays or typical development,” J. Autism Dev. Disord., vol. 44, no. 5, pp. 1117–1127, May 2014, doi: 10.1007/s10803-013-1973-x.

[7]          B. J. Ferguson, K. Dovgan, N. Takahashi, and D. Q. Beversdorf, “The Relationship Among Gastrointestinal Symptoms, Problem Behaviors, and Internalizing Symptoms in Children and Adolescents With Autism Spectrum Disorder   ,” Frontiers in Psychiatry  , vol. 10. p. 194, 2019.

[8]          M. V. Ristori et al., “Autism, Gastrointestinal Symptoms and Modulation of Gut Microbiota by Nutritional Interventions.,” Nutrients, vol. 11, no. 11, Nov. 2019, doi: 10.3390/nu11112812.

[9]          P. Gorrindo, K. C. Williams, E. B. Lee, L. S. Walker, S. G. McGrew, and P. Levitt, “Gastrointestinal dysfunction in autism: parental report, clinical evaluation, and associated factors.,” Autism Res., vol. 5, no. 2, pp. 101–108, Apr. 2012, doi: 10.1002/aur.237.

[10]        M. E. Geraghty, J. Bates-Wall, K. Ratliff-Schaub, and A. E. Lane, “Nutritional Interventions and Therapies in Autism: A Spectrum of What We Know: Part 2,” ICAN Infant, Child, Adolesc. Nutr., vol. 2, no. 2, pp. 120–133, Apr. 2010, doi: 10.1177/1941406410366848.

[11]        A. Anwar et al., “Quantitation of plasma thiamine, related metabolites and plasma protein oxidative damage markers in children with autism spectrum disorder and healthy controls,” Free Radic. Res., vol. 50, no. sup1, pp. S85–S90, Nov. 2016, doi: 10.1080/10715762.2016.1239821.

[12]        J. B. Adams and C. Holloway, “Pilot study of a moderate dose multivitamin/mineral supplement for children with  autistic spectrum disorder.,” J. Altern. Complement. Med., vol. 10, no. 6, pp. 1033–1039, Dec. 2004, doi: 10.1089/acm.2004.10.1033.

[14]        J. B. Adams, “Vitamin/Mineral Supplements for Children and Adults with Autism,” Vitam. Miner., vol. 4, no. 1, 2015, doi: 10.4172/2376-1318.1000127.

[15]        E. Pouteau et al., “Superiority of magnesium and vitamin B6 over magnesium alone on severe stress in  healthy adults with low magnesemia: A randomized, single-blind clinical trial.,” PLoS One, vol. 13, no. 12, p. e0208454, 2018, doi: 10.1371/journal.pone.0208454.

The missing 2, 13, 16 and 17 are cited already in the previous references section.

Reviewer 2 Report

This article is an interesting review focused on a clinically relevant topic such as the function / role of vitamin supplementation in autistic subjects.
I really appreciate the review work that the authors have done, but at the same time I believe that the article should be re-modulated in the concepts it wants to express.
In practice, the authors have often reported that as an effect of certain vitamins there has been an improvement in autistic children. it is precisely the concept of improvement that needs to be better explained (e.g. how have the improvements been measured? with ADOS-G, ADOS-2, CARS, CBCL?). Otherwise the improvement remains an opinion of the other authors and studies cited, but at the same time it provides incorrect and dangerous information to be disseminated on a scientific article.
Furthermore, I think the results of these studies can be harmonized with recent evidence concerning the neurochemistry of autism, as well as the introduction of the manuscript (Marotta R, et al. doi: 10.3390/brainsci10030163; Mazzone L, et al. doi: 10.3390/jcm7050102).

It is important to delete the idea that ASD is prevalent in the pediatric age, as it is the pathology of the social sphere for which it affects all ages of life, even with the possibility of diagnosis in adolescence.
I invite the Authors to be more precise also on the citations of the sources: the correct denomination is DSM-5 and not DSMV. Also as regards the historical excursus, Bleuer was the first to speak of the autistic withdrawal of schizophrenic patients, but Asperger and Kanner did not describe autism as we know it today. This concept denotes a poor knowledge of the topic. It is even stranger that the authors do not cite sources concerning current prevalence (1:68 children with an exponential increase from 1970 onwards). I therefore invite the Authors to find out better considering the ease of finding these data in the literature.

In summary I recommend a relevanti revision of the manuscript  before acceptance

Author Response

  • In practice, the authors have often reported that as an effect of certain
    vitamins there has been an improvement in autistic children. It is
    precisely the concept of improvement that needs to be better explained
    (e.g. how have the improvements been measured? with ADOS-G, ADOS-2, CARS,
    CBCL?). Otherwise the improvement remains an opinion of the other authors
    and studies cited, but at the same time it provides incorrect and
    dangerous information to be disseminated on a scientific article.

We added to each vitamin report the tools used by the researchers to measure the autism symptoms and severity as the reviewer kindly suggested.

  • Furthermore, I think the results of these studies can be harmonized with
    recent evidence concerning the neurochemistry of autism, as well as the
    introduction of the manuscript (Marotta R, et al. doi:
    3390/brainsci10030163; Mazzone L, et al. doi: 10.3390/jcm7050102).

We added these reports to our article as the reviewer proposed.

  • It is important to delete the idea that ASD is prevalent in the pediatric
    age, as it is the pathology of the social sphere for which it affects all
    ages of life, even with the possibility of diagnosis in adolescence.

Our first intention was to mention the importance of early autism identification. As the reviewer suggested we reformulated the idea of ASD prevalence among people for all the ages.

  • I invite the Authors to be more precise also on the citations of the
    sources: the correct denomination is DSM-5 and not DSMV.

We replaced DSMV with DSM-5 as the reviewer suggested.

  • Also as regards the historical excursus, Bleuer was the first to speak of the autistic
    withdrawal of schizophrenic patients, but Asperger and Kanner did not
    describe autism as we know it today.
  • It is even stranger that the authors do not cite sources
    concerning current prevalence (1:68 children with an exponential increase
    from 1970 onwards).

We added a recent report to our review about ASD prevalence where it was mentioned a new update of  number cases (1: 55 children).

Reviewer 3 Report

This can be described as a literature review but there are limitations to the studies reviewed and you tended to discuss those who showed a positive outcome. I think you should have reviewed some very good meta analysis published in Pediatrics in Nov 2019. This is not supporting non specific intervention as a treatment of ASD but suggests a potential role for some specific dietary interventions in the management of some symptoms, functions, and clinical domains in patients with ASD.

With regard to the lay out  I would suggest

a)removing the word new from line 13

b) correct the English on line 15

c) shorten or remove lines 49-57

d) I have issues with the lay out of figure 1 and I think they should be 2 figures

e) look at positioning of Table 1

References

Many of your references are quite old e.g. no 27 and I would question if these recommendations would be acceptable today. Similarly references 39, 43 especially as more recent studies have shown same results.

You need to review reference 67

Your conclusion paragraph requires further work

Author Response

  • This can be described as a literature review but there are limitations to
    the studies reviewed and you tended to discuss those who showed a positive

We wanted to highlight the importance of vitamins as a possible treatment for ASD. We reformulated and added new informations where it was necessary.

  • With regard to the layout I would suggest:

  1. a) removing the word new from line 13. Done.
  2. b) correct the English on line 15. Done.
  3. c) shorten or remove lines 49-57. Done.
  4. d) I have issues with the layout of figure 1 and I think they should be 2 figures

We changed the figure and adapted it to 2 figures now: Figure 1 for ASD characteristics and Figure 2 for vitamins effects.

  1. e) look at positioning of Table 1

It was an error but we positioned now in the right position.

  • References: Many of your references are quite old e.g. no 27 and I would question if
    these recommendations would be acceptable today. Similarly references 39,
    43 especially as more recent studies have shown same results. You need to review reference 67.

We presented in our mini-review the references available regarding those vitamins; we added new reports in the cases where it are published.  We also checked again all our cites and referenced as the reviewer kindly indicated.

We removed reference 67 because the article was retracted.

  • Your conclusion paragraph requires further work.

     We reformulated the conclusion paragraph as the reviewer indicated

Round 2

Reviewer 2 Report

The present manuscript is interesting and well-focused on relevant topic.

ASDs are a complex disease without a specific treatment and defined causes. 

In this perspective this manuscrip may contribute to the field significantly.

No relevant concerns were dectect. 

I suggest to accept it 

Reviewer 3 Report

The paper reads much better as does the flow of information

The new information added and your conclusion improves the quality of the paper.

I have no new suggestions to make

Well done on your efforts